# Molecular Characterization of Indigenous Rhizobia from Kenyan Soils Nodulating with Common Beans

**DOI:** 10.3390/ijms24119509

**Published:** 2023-05-30

**Authors:** Clabe Wekesa, Kelvin Kiprotich, Patrick Okoth, George O. Asudi, John O. Muoma, Alexandra C. U. Furch, Ralf Oelmüller

**Affiliations:** 1Matthias Schleiden Institute of Genetics, Bioinformatics and Molecular Botany, Friedrich-Schiller-University Jena, Dornburger Str. 159, 07743 Jena, Germany; clabe.wekesa@uni-jena.de (C.W.); alexandra.furch@uni-jena.de (A.C.U.F.); 2Department of Biological Sciences, Masinde Muliro University of Science and Technology, P.O. Box 190, Kakamega 50100, Kenya; kiprotichkelvin29@gmail.com (K.K.); okothpatrick@mmust.ac.ke (P.O.); jmuoma@mmust.ac.ke (J.O.M.); 3Department of Biochemistry, Microbiology and Biotechnology, Kenyatta University, P.O. Box 43844, Nairobi 00100, Kenya; asudi.george@ku.ac.ke

**Keywords:** rhizobia, symbiotic potential, pangenome, synteny, comparative genomics

## Abstract

Kenya is the seventh most prominent producer of common beans globally and the second leading producer in East Africa. However, the annual national productivity is low due to insufficient quantities of vital nutrients and nitrogen in the soils. Rhizobia are symbiotic bacteria that fix nitrogen through their interaction with leguminous plants. Nevertheless, inoculating beans with commercial rhizobia inoculants results in sparse nodulation and low nitrogen supply to the host plants because these strains are poorly adapted to the local soils. Several studies describe native rhizobia with much better symbiotic capabilities than commercial strains, but only a few have conducted field studies. This study aimed to test the competence of new rhizobia strains that we isolated from Western Kenya soils and for which the symbiotic efficiency was successfully determined in greenhouse experiments. Furthermore, we present and analyze the whole-genome sequence for a promising candidate for agricultural application, which has high nitrogen fixation features and promotes common bean yields in field studies. Plants inoculated with the rhizobial isolate S3 or with a consortium of local isolates (COMB), including S3, produced a significantly higher number of seeds and seed dry weight when compared to uninoculated control plants at two study sites. The performance of plants inoculated with commercial isolate CIAT899 was not significantly different from uninoculated plants (*p* > 0.05), indicating tight competition from native rhizobia for nodule occupancy. Pangenome analysis and the overall genome-related indices showed that S3 is a member of *R. phaseoli*. However, synteny analysis revealed significant differences in the gene order, orientation, and copy numbers between S3 and the reference *R. phaseoli.* Isolate S3 is phylogenomically similar to *R. phaseoli*. However, it has undergone significant genome rearrangements (global mutagenesis) to adapt to harsh conditions in Kenyan soils. Its high nitrogen fixation ability shows optimal adaptation to Kenyan soils, and the strain can potentially replace nitrogenous fertilizer application. We recommend that extensive fieldwork in other parts of the country over a period of five years be performed on S3 to check on how the yield changes with varying whether conditions.

## 1. Introduction

Dinitrogen gas (N_2_) is the most abundant form of biospheric nitrogen, although it is not readily available to most organisms. The transformation of N_2_ to NH_3_ (biological nitrogen fixation) mediated by some diazotrophic bacteria is the primary process by which this element is incorporated into the soil–plant system [1]. Nitrogen-fixing diazotrophs are found in various bacteria and archaea phyla, and they are either free-living, symbiotic, or associative. Rhizobia, distributed in subclasses α- and β- proteobacteria, are symbiotic bacteria that can fix nitrogen by interacting with leguminous plants. Besides their fundamental role in providing nitrogen to their host plants, numerous additional benefits to the host plants have been observed in various studies. For example, some strains of rhizobia have been implicated in phosphate solubilization [2] and the production of phytohormones such as cytokinins [3], gibberellins [4], abscisic acid [5], and indole acetic acid [6]. Furthermore, several rhizobia isolates enhance the mobilization of iron into the roots through siderophore production [7], induce systemic resistance against phytopathogens [8], or enhance tolerance to abiotic stress in their hosts [9,10].

Kenya is the seventh most prominent producer of common beans globally and the second leading producer in East Africa [11]. Beans rank second only to maize in importance as a food crop. National per-capita consumption is about 14 kg per year but can be as high as 66 kg per year in western regions. They are cultivated almost exclusively by 1.5 million smallholder farmers on about a million hectares, with about 0.6 MT/ha yields. The major producing areas for dry beans include the Rift Valley, Western, Eastern, Central, and Lake Victoria zone regions, accounting for 33%, 24%, 18%, 13%, and 20% of the national production, respectively. National consumption is approximately 755,000 MT annually against the estimated output of 600,000 MT per year [11]. These low yields may have varied reasons, including low soil quality and insufficient quantities of vital nutrients and nitrogen [12,13]. This situation has forced farmers to utilize inorganic fertilizers with known ecological consequences [14,15]. Most Kenyan soils are currently acidic, with a pH as low as 4.3 [16,17,18]. Overuse of nitrogen fertilizers causes further soil acidification [19] and dissociates insoluble Al compounds into soluble toxic Al forms. This is an increased threat to agriculture since the soluble Al concentration in Western Kenyan soil is already as high as 28 mM [20,21].

Inoculation of common beans with commercial rhizobia inoculants is one viable alternative that can be employed to circumvent these challenges. Despite the high promiscuity of common bean to rhizobia selection, its nitrogen fixation ability is far less. Even sparse nodulation or a lack of response to commercial inoculants has been reported in Western Kenya [20] and elsewhere [22]. This unfortunate situation could be related to competition from well-adapted but ineffective local rhizobia [23] to detrimental factors such as the high rate of nitrogenous fertilizer and pesticides that impair rhizobial growth and nodule formation [24,25]. Environmental factors such as low pH and heavy metals in the soil inhibit rhizobia proliferation [26,27].

The necessity for the search for a novel native rhizobial strain adapted to Kenyan soils with potential agricultural use is supported by recent comparative studies among local and commercial strains [28,29]. Most commercial inoculants used in Kenya originate from South America or the United States [20,30], such as the widely used *R. tropici* CIAT899 strain from South America. Although CIAT899 is genetically stable and tolerant to high temperatures experienced in tropical regions, it has failed to improve the yields in most Kenyan soils, probably due to low adaptation to local edaphic conditions [31]. Biofix, a rhizobia-based commercial bioinoculant for soya beans, common beans, groundnuts, and cowpea, although produced in Kenya by the Microbiological Resources Center Network (MIRCEN) in partnership with the MEA Fertilizer Ltd. and the University of Nairobi. However, the Biofix inoculants contain mainly or exclusively foreign isolates [30]. It failed to increase the yield of various legumes in Kenyan soils [28,29,32].

Several studies have established the existence of native rhizobia in Kenya with quite good symbiotic capabilities, which are far better than those of the commercial isolates when properly selected and applied in the local soils [28,32,33]. Very few such studies have been carried out in Western Kenya, and as a result, the majority of native rhizobia from Western Kenya soil, together with their symbiotic efficiency, are poorly characterized or unknown. Testing for nodulation ability and nitrogen fixation in the greenhouse under controlled bacteriology is essential for developing novel inoculant formulations. Isolates found effective in the greenhouse must then be tested in the field to check their ability to compete with local rhizobia and withstand the environmental threats of the target soils. After the successful isolation of novel rhizobia, their application in a given soil requires a three-setup procedure [34]: (1) uninoculated treatment of the plants to check for native rhizobia and their effectiveness, (2) inoculated treatment of the plants to check for the effect of the new rhizobia inoculation, and (3) uninoculated plants treated with nitrogenous fertilizer to check for the effects of N. This study aimed to test the competence of novel rhizobia strains isolated in Western Kenyan soils whose symbiotic efficiency was already confirmed in greenhouse experiments [6]. The new isolates solubilized insoluble phosphates, produced indole acetic acid in the media, and fixed more nitrogen in common beans than the commercial isolate, CIAT899, in our greenhouse studies. They grew very well in media with low pH demonstrating their adaptability in the local soils. We further characterized one of the isolates with the best nitrogen fixation ability in our field studies using whole-genome sequencing methodologies.

## 2. Results

### 2.1. Analysis of Soil Sample from Western Kenya

The soil pH at the two experimental sites was 5.20 in Kakamega and 6.08 in Busia and, therefore, officially characterized as acidic (less than 6.5) [35]. The exchangeable amounts of Al, Mn, and Cu were 1.72 Cmol/kg, 63.1 ppm, and 2.3 ppm in Kakamega and 0.95 Cmol/kg, 35.1 ppm, and 1.60 ppm in Busia soils, respectively. Most legumes, including common beans and soil microbes, are sensitive to Al concentrations < 0.5 Cmol/kg, suggesting that the local Al concentration at the two study sites may have a negative effect on the two symbionts. The Mn and Cu levels were within the plant-tolerable range, i.e., 30–500 ppm for Mn and 1–960 pmm for Cu, respectively, at pH 5–6.5 [36]. The quantities of available phosphorus in the soil at 16 ppm in Kakamega and 19.5 ppm in Busia were extremely low, as the recommended optimum range is between 30 and 50 ppm [37]. The amount of available nitrogen, 0.33 mg/g in Kakamega and 0.19 mg/g in Busia soils, was much less than the recommended amount of 2 mg/g of the soil.

Although the common beans inoculated with the isolates B3, S2, S3, and COMB (a combination of B3, S2, and S3) had higher nodule numbers than uninoculated beans, and those inoculated with CIAT899, the differences were not significant (*p* > 0.05) in both Kakamega (Figure 1A) and Busia (Figure 1B) soils. Plants supplied with nitrogenous fertilizer had fewer nodules, although the difference was insignificant compared to those inoculated with rhizobia except with S3 (*p* > 0.05) in Kakamega and both S3 and COMB in Busia soils. In both Kakamega and Busia, plants with nitrogenous fertilizer had fewer nodules than negative controls and those inoculated with rhizobia. The total dry weight of plants 28 days post-planting was not significantly different among the treatments in both study sites (Figure 1C,D).

At the completion of the experiment, i.e., the time point of harvest 70 days after planting, plants inoculated with isolate S3 or with a mixed inoculum (COMB) produced a significantly higher number of seeds and seed dry weight when compared to the uninoculated control plants at both sites (Figure 2). Except for the dry weight of seeds in Busia soils, the number and dry weight of seeds from plants inoculated with the commercial isolate CIAT899 was not significantly different from uninoculated plants (*p* > 0.05), indicating a tight competition from native rhizobia for nodule occupancy. The yield from the inoculation of the plants with isolate S3 or COMB was comparable to that from plants supplied with nitrogenous fertilizer at both sites. This demonstrates that S3 can replace nitrogenous fertilizer. However, the yield of plants inoculated with either S3 alone or COMB was not significantly different at both study sites, suggesting that the stimulatory effect in the combined inoculum was due to S3.

### 2.2. Assembly and Annotation of Isolate S3 Genome

To obtain more insight into the molecular function of S3, the complete genome was sequenced and analyzed for genomic feature composition and functional category of the annotated genes. Long read assembly resulted in the generation of five DNA fragments representing completely circularized chromosomes (4,537,630 nucleotides) and four plasmids with 446,029, 352,110, 413,456, and 1,111,454 nucleotides, respectively (RefSeq IDs: NZ_CP064931.1, NZ_CP064932.1, NZ_CP064933.1, NZ_CP064934.1, and NZ_CP064935.1). The genome was annotated to 6259 protein-coding genes, 64 RNA genes, 209 pseudogenes, 4 noncoding RNAs, 54 tRNAs, and 4 rRNAs. Three hundred sixty-eight subsystems resulted from the RAST analysis of the S3 genome (Figure 3, Appendix A). A total of 114 genes were predicted to be involved in stress responses: 21 in osmotic and salt stress, 57 in oxidative stress, 13 in detoxification, 3 in periplasmic stress, and 28 genes with protective functions which do not belong to known subsystem categories. Iron acquisition and metabolism are performed by aerobactin, a citrate-hydroxamate siderophore, and hemin. To survive environmental hostility, the S3 genome contains 64 genes involved in virulence and toxic compound resistance, i.e., copper homeostasis (18), cobalt, zinc, and cadmium resistance (10), chromium resistance (2), mercuric reduction (1), tetracycline and ribosomal protection (2), or code for beta-lactamases (2) and multidrug efflux resistance pumps (6). Some rhizobial strains secrete compounds into the host plant during infection to influence or modify the plant defense signaling pathways. Isolate S3 contains the type II secretion system (T2SS) and type IV secretion system (T4SS).

### 2.3. Pangenome Analysis

The OrthoMCL and COGtriangle programs identified 25,019 and 25,628 orthologous groups, respectively, among the genomes of 25 isolates, of which 20,487 ortho groups represented the intersection of the two programs (Figure 4). The pangenome matrix was generated from the intersection of orthologous groups of both orthoMCL and COGtriangle.

Phylogenomic analysis shows that isolate S3 forms a branch with isolates *R. phaseoli*, *R. esperanzae*, *R. etli*, *R. laguerreae*, *R. sophorae*, *R. changzhiense*, *R. leguminosarum*, *R. indigoferae*, *R. anhuiense*, and *R. acidisoli* (Figure 5). Moreover, isolate S3 has a very close genetic relationship with *R. phaseoli*, and both bacteria share the last common ancestor (LCA) with *R. esperanzae*. Further, the three isolates, S3, *R. phaseoli*, and *R. esperanzae*, share an LCA with *R. etli*.

The overall genome-related indices (ANI, AAI, dDDH, and G + C difference) were calculated to determine the relatedness between isolate S3 and ten other rhizobia strains with the closest phylogenetic relationship as indicated by the pangenome tree. Species circumscription was set at 95% for ANI [38] and AAI [39], 70% for dDDH [40], and less than 0.1% for the G + C difference [40]. The four parameters indicate that isolate S3 belongs to the *R. phaseoli* family; that is, ANI and AAI are >95%, dDDH > 70%, and the G + C difference is <0.1% (Table 1).

### 2.4. Syntenic Blocks

Synteny blocks shared between reference *R. phaseoli* and isolate S3 were identified with DNA sequences of ≥ 1000 nucleotides. One hundred and eighteen synteny blocks (Figure 6; Appendix A) were identified and annotated to 6166 genes. However, 42.7% of the annotated genes are hypothetical without known functions. Except for blocks #48 and #69 on plasmids 1 and 2, respectively, all other synteny blocks were found at different loci, and 124 (2.01%) of the blocks were inverted in S3 compared to those in *R. phaseoli*. Essential genes were still conserved even on inverted or translocated portions of the genomes. Fifty-one (0.83%) transposases were identified on the S3 genome. Transposases of the ISRel13 family were the most common and found in 14 synteny blocks, the ISRel21 family in 5 blocks, and the ISATu5 family in 3 blocks. The rest of the transposases were detected in only two or one synteny block (Appendix A). Block #25, located on plasmid 3 of the reference genome (RP3:181,596–187,250), occurred twice in the S3 genome, namely on plasmid 3 (SP3:239,618–245,269) and 4 (SP4:984,938–990,589). Although the synteny blocks on plasmid 3 of both reference and isolate S3 were in the same forward orientation, they were located approximately 52,368 nucleotides apart, while the block on plasmid 4 was in an inverted position in S3. This 5654-nucleotide block encodes five genes participating in the Type IV secretion system (T4SS, the abbreviation should be further up) (VirB4, VirB5, VirB6, VirB8, and VirB9) (Appendix A). Since the rhizobial T4SS secretes effector proteins that potentially support effective colonization of the roots, the presence of an additional synteny block in S3 may result in better nodulation effectiveness of S3 when compared to the reference strain *R. phaseoli*.

## 3. Discussion

### 3.1. Symbiotic Efficiency

We first tested for nodulation ability and nitrogen fixation in the greenhouse under controlled bacteriology. Isolates that are effective in greenhouse experiments must then be tested in the field to check their ability to compete with local isolates and withstand the environmental conditions of the target soils. To identify and determine the symbiotic efficiency of rhizobia, Maingi et al. [34] proposed the establishment of a three-setup treatment: (1) no inoculation to check on the availability of native rhizobia and their effectiveness, (2) inoculation to test the effect of rhizobia inoculation, and (3) a comparison of uninoculated plants treated with nitrogenous fertilizer to determine whether rhizobia inoculation can replace nitrogenous fertilizer in a particular soil.

We tested whether the performance of bean plants (measured as nodule number, plant biomass, number, and dry weight of the seeds per plant) was promoted on local soil when exposed to either the novel isolates B3 or S2 and S3 or the commercial strain CIAT899. The nodule number was used to assess the nodulation efficiency of B3, S2, and S3 compared to CIAT899 and to the indigenous rhizobia in the soil. As indirect parameters, the plant biomass and the number and dry mass of the produced seeds were used to compare the nitrogen fixation efficiency among the different strains.

There was no significant difference in nodule number on the inoculated and uninoculated plants, showing that the application of B3, S2, S3, and CAI899 has no advantage over the local bacteria nodulating common beans in the local soil. Consequently, none of the applied bacteria promoted the biomass of the plants within 28 days after planting. Even N fertilization did not promote the biomass of the plants, indicating that rhizobia, either exogenously applied or indigenous in the soil or seeds, have either no or only a minor effect on biomass production. Apparently, the plants obtain sufficient nutrients from the soil or seed during the first 28 days after planting, consistent with previous findings [28,34]. However, 70 days after planting, plants inoculated with either S3 alone or a mixed inoculum containing S3 had a significantly higher seed number and dry weights than the uninoculated controls and plants inoculated with the commercial rhizobia CIAT899. The lower seed dry weight of the uninoculated plants is most likely caused by N limitation since N fertilization also promotes the seed weights. A beneficial effect was also observed for B3 and S2, but the effects were less efficient than for S3. These results suggest that S3 might be an efficient nodulator with agricultural relevance. Consistent with results from similar studies [28], the beneficial effect of S3 was not further stimulated when the strain was applied in combination with B3 and S2.

### 3.2. Competition Strategies for Nodule Occupancy

A novel rhizobium must effectively compete with other rhizobia and non-rhizobia in the rhizosphere to enter a symbiotic nitrogen-fixing relationship with a host plant. Competition implies preventing the occupation of the niche by different bacterial strains. Rhizobia achieves this either indirectly by exploitative utilization of a common limited nutrient or directly by interfering with the growth of other bacteria. Genome analysis of isolates S3 revealed the presence of siderophore aerobactin biosynthesis genes primarily found in pathogenic bacteria, such as *Salmonella*, *E. coli*, *Shigella* and *Klebsiella* [41]. Aerobactin siderophores play essential roles under iron-limited conditions and oxidative stress resistance, ROS removal, biofilm formation, and virulence in the food-borne enteric pathogen *Yersinia pseudotuberculosis* [42]. Therefore, the siderophore produced by S3 may have several functions: sequestering iron for its use and thereby limiting the growth of competitors and acting as an antibacterial agent to restrict the survival of other bacteria/rhizobia in the rhizosphere or nodule. Furthermore, the participation of aerobactin siderophores in biofilm formation forms a barrier that protects rhizobia from antimicrobials released by other bacteria [43,44]. Furthermore, biofilm formation protects bacteria against a variety of abiotic stresses, such as drought [45], extreme pHs [44], and salt stress [46].

Notably, the genome of S3 contains several genes involved in antibiotic tolerance. For example, *creA* is responsible for colicin two tolerance [47]. The genes for ribosome-protecting tetracycline resistance and translation elongation factor G might protect the rhizobia against tetracyclines in their environment. Other antibiotic-resistant genes present on the S3 genome code for beta-lactamase hydrolases for beta-lactamase destruction, multidrug resistance pumps, including the RND family MDR pump, the MATE family MDR efflux pump, an ABC efflux transport system, and a Major facilitator superfamily multidrug-efflux transporter, which actively exports antibiotics from the bacterial cells [48].

### 3.3. Genomic Strategies for Toxic Metal Tolerance

Although Cu is an essential element in cell physiology as part of prosthetic centers of oxidases and cytochrome biogenesis proteins, a high concentration can harm rhizobia cells [49]. Therefore, an accurate balance of Cu is crucial for cell homeostasis. Isolates S3 contain genes for a Cu efflux regulator (CueR), Cu-translocating P-type ATPase (CtpA), mult-Cu oxidase, Cu resistance protein G (CopG), and the Cu homeostasis protein CutE. CueR controls the transcription of two Cu-homeostasis genes. CopA is a Cu^+^-transporting P-type ATPase pump, and CueO is a Cu oxidase for detoxification [50,51]. Telianidis et al. (2013) showed that CtpA catalyzes ATP-dependent Cu transport across cell membranes, which essential cuproenzymes require. It removes excess cellular Cu to prevent Cu toxicity. Multicopper oxidases participate in Cu tolerance in *B. melitensis* [52] and *E. coli* [53] since deleting genes responsible for these proteins increases the susceptibility of the bacteria to Cu toxicity. CopG, widespread in Gram-negative bacteria, plays a role in interconversion between Cu(I) and Cu(II) to minimize toxic effects and facilitate its export by the Cus RND transporter efflux system [54].

The divalent ions of Co, Zn, and Ni as trace elements at nanomolar concentrations are essential nutrients for bacteria. However, at micro- or millimolar concentrations, Co^2+^, Zn^2+^, Ni^2+^, and perhaps some other ions without nutritional roles, such as Cd^2+^, are toxic [55]. The genome of S3 has a *czc* gene that codes for a Co-Zn-Cd resistance protein, the Cd(II)/Pb(II)–responsive transcriptional regulator TRCd, and the heavy metal resistance transcriptional regulator HmrR. Both are involved in the efflux of these metal ions from the cell, enabling bacteria to survive in the contaminated environment [55,56].

Another gene located in the genome of S3 is *chrA* involved in Cr resistance. Cr compounds are severe environmental contaminants. Chromate (CrO_4_^2−^) exerts diverse toxic effects on bacteria, including inhibition of sulfate transport, as well as DNA and protein damage after intracellular reduction of Cr(VI) to Cr(III). Resistance to Cr compounds can be established by membrane transporters encoded by the *chrA* genes, which directly mediate the efflux of chromate ions from the cell’s cytoplasm and detoxification reactions whose transcription is induced by chromate [57]. These efflux systems in the isolate S3 genome may be responsible for its survival in toxic environments and may also be involved in detoxifying other metallic toxins not considered in this study.

### 3.4. Genomic Adaptations to Osmotolerance

Prolonged periods of drought or insufficient water, especially in the tropics, can increase salt concentrations in the soil. A high solute concentration in the environment reduces the amount of water readily available to the soil bacteria. Some prokaryotes can maintain water availability in these environments by increasing their cellular solute concentrations. The genome of S3 contains choline uptake and dehydrogenation pathways genes, genes for the osmoprotectant ABC transporter system YehZYXW, a glucan biosynthesis protein system, and the biosynthesis of aquaporin Z channels. Glycine betaine is a very efficient osmolyte found in many bacterial species and accumulates at high cytoplasmic concentrations in response to osmotic stress. It is synthesized from choline transported into the cell [58] or from choline sulfate through choline-O-sulfatase (BetC) [59]. A FADH-dependent choline dehydrogenase (BetA) catalyzes the formation of an intermediate, glycine betaine aldehyde which is then converted into glycine betaine through the NADH-dependent glycine betaine aldehyde dehydrogenase (BetB) [60]. Genes for choline uptake and dehydrogenation (*betA*, *betB*, and *betC*) are present in the genome of S3.

Furthermore, the S3 genome also contains genes for sarcosine oxidases that catalyze the oxidative demethylation of sarcosine to yield glycine. The *yehZYXW* operon encodes the “ABC transporter family’s putative osmoprotectant uptake system.” The transporter system is induced by osmolarity upon cell entry into the stationary phase [61]. However, the substrate of the ABC transporter remains unknown. It was shown that the transporter is not involved in glycine betaine, choline, or choline-O-sulphate transport [62]. Osmoregulated periplasmic glucans (OPGs) are oligosaccharides of 5–14 D-glucose units with β-glycosidic bonds. The OPG concentration in the periplasm increases in response to decreased environmental osmolarity [63]. Aquaporin Z from the *aqpZ* gene mediates water’s rapid entry or exit in response to abrupt changes in osmolarity. It is involved in osmoregulation and cell turgor maintenance during volume expansion in rapidly growing cells [64].

### 3.5. Synteny Analysis

Phylogenomic analysis revealed that S3 is closely related to *R. phaseoli*. ANI, AAI, dDDH, and G + C difference data confirmed that isolate S3 is *R. phaseoli*. However, these methodologies are based only on local mutations (substitution or point mutation, deletion, and insertion of a single nucleotide), leading to sequence divergence between the homologous genes in separate strains. Global mutations (genome rearrangements) include inversions, translocations, transposition, and duplication events and result from different biological mechanisms such as recombination, DNA repair, and replication. Genome rearrangements can create novel genes by splitting or fusing existing ones and can also change the position of genes within a chromosome or from one chromosome to another [65]. Except for two synteny blocks, the rest have undergone rearrangements on the genome of the Kenyan isolate S3 when compared to reference *R. phaseoli*. The hypothesis for selection for niche adaptation (SNAP) proposes that rearrangements in the chromosomal gene order are selected indirectly as a consequence of selection acting on organisms to adapt to changing or new environmental niches [65]. For example, a 1.66 Mb DNA portion was duplicated on the *Salmonella typhi* chromosome when the bacterium was adapted to grow on malate as the sole carbon source, which enabled an increased copy number of a single gene, *dctA* that is involved in the uptake of malate [66].

In our study, a 52.368 kb DNA fragment on reference *R. phaseoli* plasmid 3 (symbiotic plasmid) was not only found to have translocated 58,022 positions and in reverse orientation on the symbiotic plasmid, but also a duplicated copy was detected on a non-symbiotic plasmid in the Kenyan isolate S3. This section carries five genes, VirB4, VirB5, VirB6, VirB8, and VirB9, that participate in the type IV secretion system, whose genes are homologous to the Vir system of *A. tumefaciens*. Hubbert et al. (2007) found that T4SS affects the rhizobia host specificity on symbiosis, probably due to the secretion of two effector proteins, Msi059 and Msi061, into the host cells. The expression of T4SS genes is under the control of the NodD gene product, a rhizobium protein that induces the nodulation of rhizobia with its host [67]. T4SS system is also speculated to be involved in suppressing plant immune response to rhizobia enabling successful nodulation [68,69]. More T4SS gene copies in the Kenyan isolate most likely enable effective competition of this isolate for nodule occupancy.

## 4. Materials and Methods

### 4.1. Study Site and Soil Sample Analysis

The symbiotic efficiency of three novel isolates, called B3, S2, and S3, initially characterized in greenhouse experiments [6], were determined at Masinde Muliro University of Science and Technology agricultural farms (0.2827° N, 34.7519° E) in Kakamega County and Nambale (0.4493° N, 34.2519° E), Busia County, Kenya. Kakamega lies about 60 km from Lake Victoria (the world’s second-largest freshwater lake), while Busia lies on the border between Kenya and Uganda. The pH was measured with the Digital pH meter (Avi Scientific India, Maharashtra, India) in a 1:2.5 soil water suspension. The minerals P, Cu, and Zn were extracted with a Mehlich-1 reagent [70], and their quantities were estimated by the inductively coupled plasma optical emission spectroscopy [71]. Exchangeable Al and available nitrogen (ammonium-N and nitrate-N) were extracted with 1 M KCl and quantified using a LACHAT Flow Injection Autoanalyzer (Hach, CO, USA) [72].

### 4.2. Field Experiment

Field studies were conducted with the common bean (Rosecocco variety) (Kenya Seed Company LTD, Kitale, Kenya) as the host legume. Rosecocco are most popular been in the Kenyan market, are tolerant to common bean diseases, and usually has high yields. We tested whether applying the novel rhizobial isolates B3, S2, and S3 (either alone or as a mixture) or the commercial strain CIAT899 influences the plants’ performance and yield, thereby keeping in mind that the plants are already exposed to the indigenous rhizobia. We measured nodulation (nodules/plant) and growth (dry weight/plant) of the plants after 28 days and the total seed yield (dry seed weight/plant) at the harvest time point. The experimental setup was in a randomized complete block design (RCBD) with three replications. Plants were treated with the isolates B3, S2, or S3 individually or combined with all three strains (COMB) or the commercial isolate *R. tropici*, CIAT899. Non-inoculated and nitrogen-fertilized samples were incorporated as negative (Neg), and positive (Pos) controls. The plant spacing used was 70 cm by 40 cm. Each plot measured 2 × 2 m, and a spacing of 1 m between each plot was left to reduce inter-plot interference. Seeds for inoculation were immersed in a 5% sucrose solution to enhance their adhesion to inoculant carrier materials. Sucrose-coated seeds were thoroughly mixed with sterile filter mud (100 g seeds per 15 kg mud, cf. [34] before planting four seeds per hole during the start of the short rain season (July–September, 2021). The filter mud (a waste product from sugar cane mills) had been previously soaked in an OD = 0.5 nm YMB rhizobial broth (1 L/15 Kg mud) containing B3, S2, S3, or CIAT899. COMB inoculant was prepared by combining a third of every B3, S2, and S3 inoculant.

Plants were weeded and thinned to two after ten days, and the second weeding was performed 20 days later. After 28 days, six plants per treatment in each plot were randomly sampled to record the number of nodules and total dry weight per plant. After maturity (70 days after planting), six plants in each plot per treatment were randomly harvested, and the total dry weight of seeds per plant was recorded. Dry weights were obtained after drying plant materials in the oven at 70° C for 48 h to constant mass. The obtained data were subjected to one-way ANOVA followed by pairwise mean separation with Tukey’s HSD test in Python v3.9.

### 4.3. Whole-Genome Sequencing, Assembly and Annotation

The high-molecular-weight DNA was extracted from the 4-day-old isolate S3 cultures using MagAttract^®^ HMW DNA Kit (QIAGEN GmbH, Düsseldorf, Germany) which enables purification of high-molecular-weight DNA using a simple, magnetic-bead-based protocol. The single-molecule real-time (SMRT) sequel sequencing technology [73] from PacBio (Pacific Biosciences, Menlo Park, CA, USA) was used to create high-quality long-read datasets of isolate S3 at Novogene (London, United Kingdom). The quality filtering was performed automatically during assembly using the SMRT Portal P-filter module and the Hierarchical Genome Assembly Process 3 (HGAP3) pipeline. Complete genome assembly was performed de novo with Flye v. 2.8.1 [74] software using default settings. Gene annotation was performed by the Prokaryotic Genome Annotation Pipeline (PGAP) [75] and subsystem classification in the Rapid Annotation using the Subsystem Technology tool kit (RASTtk) pipeline [76].

### 4.4. Pangenome and Synteny Analyses

Twenty-three GenBank files for completely sequenced rhizobia and one *Agrobacterium tumefaciens* were retrieved from the National Center for Biotechnology Information (NCBI) (Table 1). Together with the genomic sequence of isolate S3, they were used to cluster the genes into their respective cluster of orthologous groups (COGs) using GET_HOMOLOGUES v3.3.2 [77], which utilizes COGtriangle [78] and OrthoMCL [79]. The Perl script compare_clusters.pl in GET_HOMOLOGUES was used to generate the pangenome matrix representing the intersection between clusters generated by COGtriangle and OrthoMCL. The pangenome matrix was used to create the maximum parsimony pangenome tree with IQ-TREE v1.6.12 in GET_PHYLOMARKERS v2.0.1 [80], and the tree was visualized in the FigTree v1.4.4 [81] tree viewer.

Species delimitation was performed on ten reference isolates with very close phylogenomic relationship to isolate S3, as guided by the pangenome tree. The overall genome-related indices (OGRIs), including average nucleotide identity (ANI), average amino acid identity (AAI), digital DNA-DNA hybridization (dDDH), and the difference in guanine and cytosine (GC) content were calculated with ANI, which was computed with fastANI v1.33 [38], AAI with EzAAI v1.2.2 [82], dDDH and difference in GC content with the genome-to-genome distance calculator (GGDC) v3.0 [40,83]. Synteny analysis was performed using Sibelia v3.0.7 [84], and the results were visualized as chords with Circos v0.69-9 [85]. Prokka v1.13.4 [86] annotated the resulting synteny blocks to their respective genes.

## 5. Conclusions

In conclusion, we have shown through field experiments that Kenyan isolates are well adapted to the local soils, and S3 has the potential to replace nitrogenous fertilizer and foreign rhizobia application due to its high symbiotic efficiency. Although phylogenetically similar to *R. phaseoli*, it has undergone significant genome rearrangements (global mutagenesis) to adapt to the Kenyan soils. We recommend that extensive fieldwork in other parts of the country over a period of five years be performed on S3 to check on how the yield changes with varying whether conditions.

## Figures and Tables

**Figure 1 ijms-24-09509-f001:**
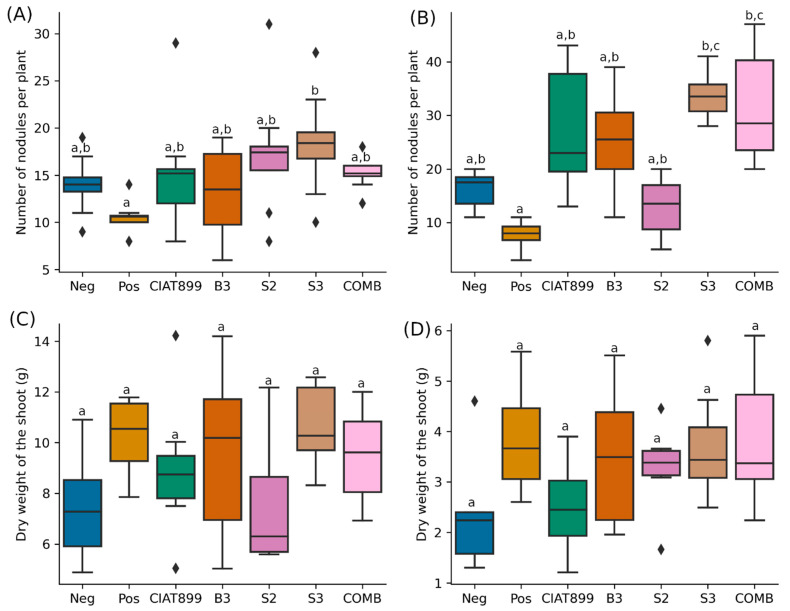
Number of nodules per plant in Kakamega (**A**) and Busia (**B**) soils and dry weight of the shoots from Kakamega (**C**) and Busia (**D**) soils 28 days before planting. Diamonds represent outliers in the data while different lower-case letters above bars indicate significant differences (*p* ≤ 0.05).

**Figure 2 ijms-24-09509-f002:**
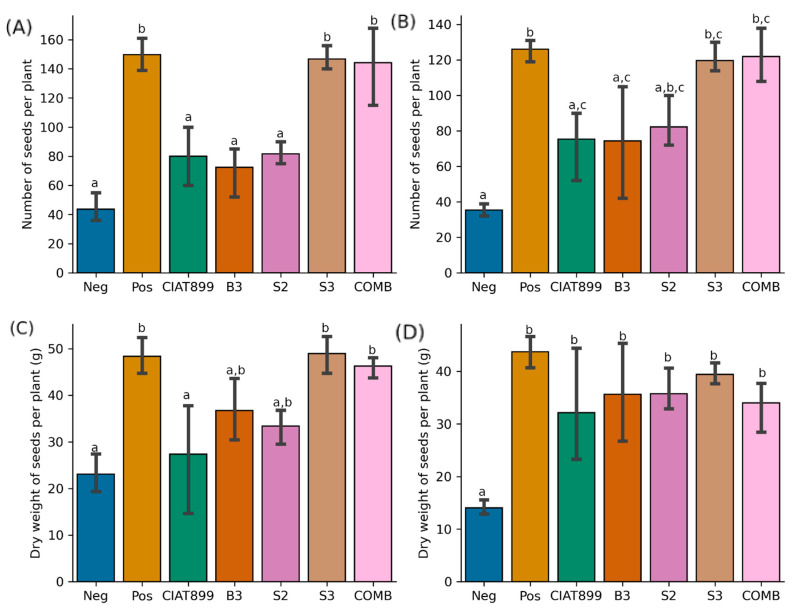
Yield of common beans in terms of the number of seeds per plant in Kakamega (**A**) and Busia (**B**) and dry weight of seeds per plant in Kakamega (**C**) and Busia (**D**). Diamonds represent outliers in the data while different lower-case letters above bars indicate significant differences (*p* ≤ 0.05).

**Figure 3 ijms-24-09509-f003:**
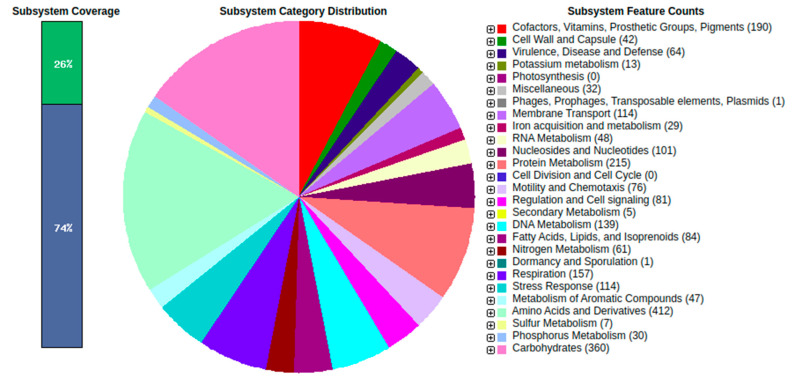
Subsystem classification of the annotated isolate S3 genes.

**Figure 4 ijms-24-09509-f004:**
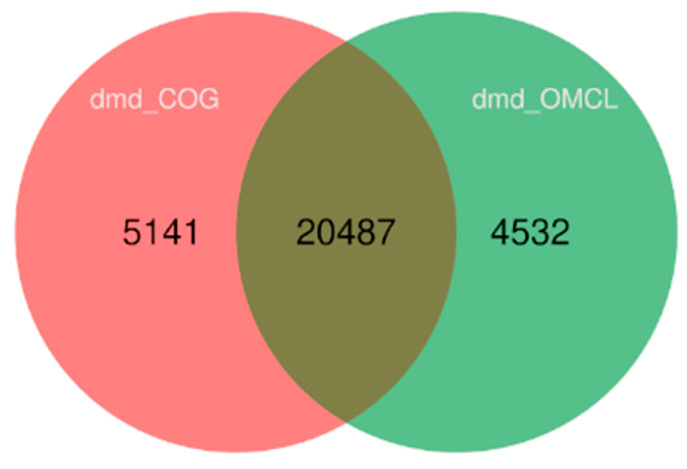
Venn diagram showing the intersection of orthologous groups resulting from COGtriangle and orthoMCL outputs.

**Figure 5 ijms-24-09509-f005:**
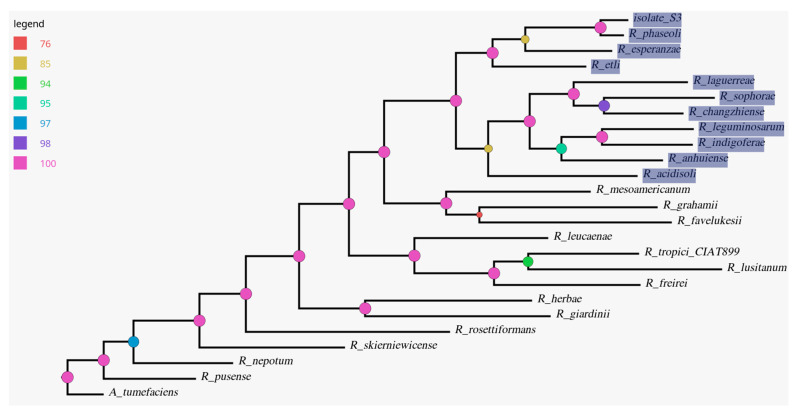
The most parsimonious pangenome tree. The nodes are colored in relation to the legend and represent standard bootstrap support values computed by seqboot from the PHYLIP package. Highlighted taxa represent isolate S3’s closest neighbors.

**Figure 6 ijms-24-09509-f006:**
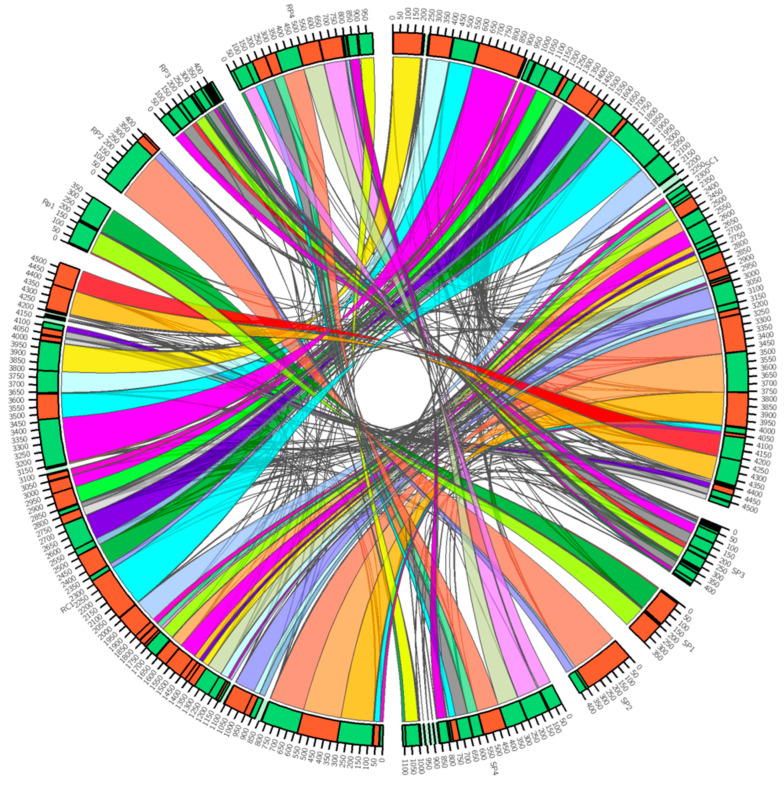
Synteny blocks (regions of chromosomes or plasmids between genomes that share a common order of homologous genes believed to have been derived from a common ancestor) between *R. phaseoli* and isolate S3SC1 and RC1 represents, respectively, represents chromosomes from isolates S3 and *R. phaseoli*, SP1–4 and RP1–4 represents plasmids from S3 and *R. phaseoli*, respectively. Similar color bands point to similar blocks representing synteny blocks in identified between S3 and *R. phaseoli*.

**Table 1 ijms-24-09509-t001:** Average nucleotide identity (ANI), average amino acid identity (AAI), digital DNA-DNA hybridization (dDDH), and G + C difference between isolate S3 and ten other reference rhizobia genomes from NCBI.

References	ANI	AAI	dDDH	G + C Difference
*R. phaseoli*	99.7635	99.74	96.7	0.03
*R. esperanzae*	90.4056	92.32	69.2	0.20
*R. etli*	89.5756	91.62	60.0	0.27
*R. acidisoli*	88.9017	90.52	54.7	0.27
*R. anhuiense*	87.7826	89.11	48.1	0.34
*R. changzhiense*	87.5833	89.02	44.3	0.22
*R. sophorae*	87.5225	88.70	43.1	0.48
*R. indigoferae*	87.5195	89.04	45.6	0.64
*R. leguminosarum*	87.5068	89.04	44.7	0.64
*R. laguerreae*	87.4644	88.38	43.6	0.42

## Data Availability

The datasets generated and analyzed during the current study are available in the NCBI RefSeq repository, https://ftp.ncbi.nlm.nih.gov/genomes/all/GCF/012/241/395/GCF_012241395.2_ASM1224139v2/ (accessed on 25 April 2023).

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
