# Peer review of "Molecular Characterization of Indigenous Rhizobia from Kenyan Soils Nodulating with Common Beans"

_ijms, 2023, doi:10.3390/ijms24119509_

Round 1

Reviewer 1 Report

The paper is well written and highly informative. The paper is well written with beautiful graphical illustrations. I highly recommend this paper for publication. Minor comments are mentioned below.

Abstract: Repeated use of ‘however’

Author Response

Abstract: Repeated use of ‘however’

Replaced the second occurence of "however" with "nevertheless"

Reviewer 2 Report

The article is devoted to the selection and testing of native rhizobia strains isolated from native Western Kenya soils. Their symbiotic efficiency was first successfully determined in greenhouse experiments. Then they were tested in the field conditions. Rhizobial isolate S3 was selected as the best one for inoculation and increase the yield of soybeans. It was discovered that this rhizobia strain has undergone important genome rearrangements to adapt to harsh conditions of Kenyan soils. Obtained results show the importance and necessity of native rhizobia evaluation in different regions of legumes production to select the best genotypes, adapted to particular growing conditions.

The article can be published in present form.

Author Response

There was no comment from the reviewer

Reviewer 3 Report

Dear authors!

An article submitted for review is written at a high scientific level using modern molecular genetic techniques and bioinformatics methods.

Questions to the authors:

1. According to the information indicated in the article, you conducted a field experiment for only one season, in 2021. I consider this a big minus of the work. I would like to note that in order to form confident conclusions, it is necessary to conduct field trials for 5 years. The authors need to describe how variable the climate is in the region where the field experiments were carried out and explain why the results of only one-year tests are not considered sufficient.

2. Please indicate the exact number of plants in each option that you used in the field experiment.

3. Indicate where the bean seeds were obtained from. Explain why this variety was chosen for the experiments. What is the bean variety used in terms of characteristics (resistance to biotic and abiotic factors)?

4. It is necessary to add a description of fig. 6. In its present form, it is understandable only to specialists in this narrow field.

5. Of the 86 sources in the list of references, only 16 sources are for 2020-2022. The list of references needs to be updated.

6. Names of species in Latin are often not italicized, which is incorrect.

I believe that the article can be successfully published in the IJMS journal after a little revision.

Respectfully Yours, reviewer.

May 15, 20203

Author Response

1. According to the information indicated in the article, you conducted a field experiment for only one season, in 2021. I consider this a big minus of the work. I would like to note that in order to form confident conclusions, it is necessary to conduct field trials for 5 years. The authors need to describe how variable the climate is in the region where the field experiments were carried out and explain why the results of only one-year tests are not considered sufficient.

We agree with the reviewer that we should have taken 5 years to make the study conclusive. However, this was a doctoral research project with very limited time. However, we planning to perform a thorough fieldwork on the isolates in various many more plots over a period of 5 years to determine if the isolate is fit for commercialization. To capture this, we have changed the statement, “We recommend that extensive fieldwork in other parts of the country be performed on S3 to enable its commercialization.” to “We recommend that extensive fieldwork in other parts of the country over a period of five years be performed on S3 to check on how the yield changes with varying whether conditions” in the abstract (line 33-36) and conclusion (line 504-506).

2. Please indicate the exact number of plants in each option that you used in the field experiment.

We already included it on lines (lines 454-456)

3. Indicate where the bean seeds were obtained from. Explain why this variety was chosen for the experiments. What is the bean variety used in terms of characteristics (resistance to biotic and abiotic factors)?

Corrected (line 433-435)

4. It is necessary to add a description of fig. 6. In its present form, it is understandable only to specialists in this narrow field.

Corrected

5. Of the 86 sources in the list of references, only 16 sources are for 2020-2022. The list of references needs to be updated.

Unfortunately, we could not find all the information that we needed in only very recent papers and so we had no choice but to use these slightly old papers.

6. Names of species in Latin are often not italicized, which is incorrect.

corrected